# How We Manage Myelofibrosis Candidates for Allogeneic Stem Cell Transplantation

**DOI:** 10.3390/cells11030553

**Published:** 2022-02-05

**Authors:** Nicola Polverelli, Mirko Farina, Mariella D’Adda, Enrico Damiani, Luigi Grazioli, Alessandro Leoni, Michele Malagola, Simona Bernardi, Domenico Russo

**Affiliations:** 1Unit of Blood Diseases and Bone Marrow Transplantation, Cell Therapies and Hematology Research Program, Department of Clinical and Experimental Sciences, University of Brescia, ASST Spedali Civili di Brescia, P.le Spedali Civili 1, 25123 Brescia, BS, Italy; mirkfar@gmail.com (M.F.); alessandro.leoni@unibs.it (A.L.); michele.malagola@unibs.it (M.M.); simona.bernardi@unibs.it (S.B.); domenico.russo@unibs.it (D.R.); 2Hematology Division, Department of Oncology, ASST Spedali Civili di Brescia, P.le Spedali Civili 1, 25123 Brescia, BS, Italy; marielladadda@libero.it; 32nd Division of General Surgery, Department of Medical & Surgical Sciences, ASST Spedali Civili di Brescia, P.le Spedali Civili 1, 25123 Brescia, BS, Italy; enricoandreadamiani@libero.it; 4Department of Radiology, ASST Spedali Civili di Brescia, P.le Spedali Civili 1, 25123 Brescia, BS, Italy; luigi.grazioli@asst-spedalicivili.it; 5CREA Laboratory (Centro di Ricerca Emato-Oncologica AIL), ASST Spedali Civili di Brescia, P.le Spedali Civili 1, 25123 Brescia, BS, Italy

**Keywords:** myelofibrosis, allogeneic stem cell transplantation, bone marrow transplantation, ruxolitinib, splenectomy, JAK-inhibitors, iron overload, deferasirox

## Abstract

Moving from indication to transplantation is a critical process in myelofibrosis. Most of guidelines specifically focus on either myelofibrosis disease or transplant procedure, and, currently, no distinct indication for the management of MF candidates to transplant is available. Nevertheless, this period of time is crucial for the transplant outcome because engraftment, non-relapse mortality, and relapse incidence are greatly dependent upon the pre-transplant management. Based on these premises, in this review, we will go through the path of identification of the MF patients suitable for a transplant, by using disease-specific prognostic scores, and the evaluation of eligibility for a transplant, based on performance, comorbidity, and other combined tools. Then, we will focus on the process of donor and conditioning regimens’ choice. The pre-transplant management of splenomegaly and constitutional symptoms, cytopenias, iron overload and transplant timing will be comprehensively discussed. The principal aim of this review is, therefore, to give a practical guidance for managing MF patients who are potential candidates for allo-HCT.

## 1. Introduction

Allogeneic hematopoietic stem cell transplantation (allo-HCT) still represents the only curative option for patients with myelofibrosis (MF), a myeloproliferative neoplasm characterized by splenomegaly, constitutional symptoms, anemia, and a natural progression to acute leukemia [1]. The median age at diagnosis (roughly 60 years) of MF and the significant transplant-related morbidity and mortality limited, in the past, the use of allo-HCT only to a minority of patients. However, during the last few years, a series of advancements led to a change of this scenario. Novel and less toxic conditioning platforms, as well as a better HLA donor selection and GVHD and anti-infective therapies, greatly improved the feasibility and safety of allo-HCT. All these advances led to extending the indication for a transplant to a larger number of elderly patients affected by hematological neoplasms, including MF [2].

The increasing proportion of older allo-HCT candidates was recently confirmed in an Italian multicenter study that analyzed the GITMO allogeneic transplant activity in elderly individuals (>60 years) between 2000 and 2017. In that experience, the median age at transplant gradually raised over time, and more than 50% of all transplants were performed in the last 5-year period [3]. In parallel, the use of unrelated and mismatched related donors also increased, and transplant procedure was progressively extended to less fit patients, thanks to an increased use of reduced intensity conditioning regimens. 

In addition to transplant procedures’ improvements, JAK-inhibitors have significantly ameliorated the clinical conditions of MF patients, thus allowing to consider allo-HCT in an increasing number of individuals [4,5,6,7,8]. Accordingly, the number of transplant procedures for MF steadily increased over time, as confirmed by several reports of the European Society for Blood and Marrow Transplantation (EBMT). Specifically, in a Europe-wide analysis including 4142 patients submitted to transplant for MF between 1995 and 2018 across 278 Centers, only 389 patients were transplanted before 2006; conversely, 1695 patients belonged to the earlier transplant era (2015–2018) [9]. The numbers of transplant procedures exponentially rose after 2012, when treatment with ruxolitinib was widely available in Europe. Whether the adoption of JAK-inhibitors utilized prior to transplant have improved the outcomes remains matter of debate and points out the importance of the pre-transplant management of MF patients. In other words, the availability of effective medical treatments and better transplant outcomes raises the need for a proper patient selection and management.

The MF transplant indication, the judgement of patient eligibility for a transplant, the choice of donor/stem cell source, and conditioning, as well as the clinical management of MF patients waiting for a transplant, will be addressed in the present review. The aim is to give a practical guide for dealing with the process from allo-HCT indication to transplantation in MF.

## 2. Transplant Indication

Clinical prognostic scoring systems play a pivotal role for guiding selection of MF patient who may benefit from allo-HCT. The most frequently used scores are: (1) the International Prognostic Scoring System (IPSS) [10], which estimates survival at the time of MF diagnosis; (2) the Dynamic IPSS (DIPSS) [11], utilizing the same five factors of IPSS, but applicable at any stage during the disease course; (3) and the DIPSS-plus, which considers three additional adverse factors (transfusion dependency, thrombocytopenia <100 × 10^9^/L, and unfavorable cytogenetics) [12,13]. 

Currently, the European LeukemiaNet/EBMT expert consensus, published in 2015, proposed that patients with intermediate-2- or high-risk disease according to the IPSS, DIPSS, or DIPSS-plus and age <70 years should be considered potential candidates for allo-HCT [14]. Conversely, the indications for transplantation for intermediate-1-risk patients is still debated [15]. The aforementioned consensus suggested that patients with intermediate-1-risk disease and age <65 years should be considered as candidates if they present with either transfusion-dependent anemia, or a significant percentage of peripheral blasts (>2%), or adverse (as defined by the DIPSS-plus classification) cytogenetics [14]. These recommendations were released in the light of a previous study that reported the comparative effect of a transplant versus a non-transplant approach across several European and American Hematological Centers. By merging the original DIPSS dataset with 255 MF patients treated with conventional therapy and a cohort of 188 transplanted MF subjects, the authors found a net benefit for transplantation only among intermediate-2 or high risk DIPSS patients. Regarding intermediate-1 DIPSS patients, survival curves did not significantly differ in the two groups [15]. Similar results were confirmed in a recent CIBMTR analysis [16]. Herein, a survival advantage was seen for the transplant in the DIPSS intermediate 1 group; however, this improvement was apparent only one year after transplant, due to a worse NRM in the peri-transplant period. Taken together, these data seem to further confirm the validity of ELN/EBMT 2015 indications, where intermediate 1 DIPSS patients should receive individual counseling. 

However, such scores have been proven to poorly predict patients’ outcome for patients with secondary myelofibrosis (sMF; post-Polycythemia Vera (PPV-MF) or post-Essential Thrombocythemia myelofibrosis (PET-MF)), as they present a better survival compared to primary myelofibrosis [17,18]. For those patients with sMF, the so-called Myelofibrosis SECondary to PV and ET-Prognostic Model (MYSEC-PM) was developed and documented better prognostic ability [19,20]. In addition, its superior predictive role was confirmed in the allo-HCT setting in 159 sMF allo-HCT patients [21].

Subsequently, several somatic mutations have been found in patients with MF, and some of them have been shown to have an impact on survival (OS) and on the risk of disease progression and blastic transformation [22,23,24]. Triple negative patients (e.g., lacking of a detectable JAK2, CALR, or MPL mutation) present a poor outcome (median survival of 3.2 years), as well as high molecular risk patients, which are conventionally defined by the presence of at least one of the EZH2, ASXL1, IDH1/2, and SRSF2 mutations, and who are associated with both worse overall and leukemia-free survival [25]. In addition, U2AF1Q157 mutation was recently confirmed as a detrimental risk factor for disease progression and survival [26]. Considering the poor outcome of these patients, some authors questioned whether earlier transplantation should be considered for those who have ‘triple negative’ disease or high molecular risk profile, even if they belong to lower risk classes according to standard prognostic scores [27]. Actually, a study on behalf MPD research consortium documented that allo-HCT was able to overcome the prognostic value of several of these mutations in a cohort of 101 patients, supporting the value of early transplantation in such a high-risk population [28].

In order to integrate the modern molecular information, the mutation-enhanced IPSS “MIPSS70” and “MIPSS70-plus” (including cytogenetics) scoring systems have been developed as decisional tools for transplant indication in patients less than 70 years old. Significant risk factors for overall survival were leukocyte count >25 × 10^9^/L, platelet count <100 × 10^9^/L, presence of high molecular risk mutations, hemoglobin <100 g/L, peripheral blood blasts ≥2%, constitutional symptoms, high molecular risk category, fibrosis grade >2, and absence of CALR type-1-like mutations and adverse cytogenetics (the latter in MIPSS70-plus only). Recent updates to ‘MIPSS70-plus version 2′ occurred with the recognition of U2AF1Q157 and new sex- and severity-adjusted hemoglobin thresholds [29]. Importantly, these scores incorporate current molecular data and up-to-date WHO 2016 disease classification and can aid decisions regarding allo-HCT. Finally, a genetically inspired prognostic score system (GIPSS) exclusively based on genetic markers is available [30].

Taking into account the large number of available prognostic tools, a question arises: What to use?

Currently, it is unclear which score is more accurate in defining the indication for a transplant, particularly when a patient falls into categories with markedly different survival expectations [31]. In daily practice, for example, often, a patient with MF may present different IPSS and MIPSS-70 scores with significant difference in OS. This kind of discrepancy between risk models may occur in up to 50% of patients with sMF [20], resulting in significative challenges in transplant indication. The use of MYSEC-PM seems to be more accurate in sMF prognostication, while, in primary MF, no clear evidence of superior efficacy between different prognostic scores is demonstrated in the allo-HCT setting; therefore, each Center should follow its institutional policy (e.g., unavailability of extended molecular analysis, medullary punctio sicca with failed cytogenetics). Table 1 summarizes the available prognostic scores, highlighting those categories of patients with shorter survival, suitable for transplantation.

## 3. Patient Selection

Once defining the indication for a transplant, the evaluation of patients’ eligibility for allo-HCT is crucial for the selection of appropriate candidates for such an intensive procedure. In fact, non-relapse mortality represents one of the major limits for transplant success, particularly in MF. The selection of MF patients may represent a crucial point. Currently, we have several tools useful for the evaluation of patients’ eligibility to transplant. However, only a minority of them has been specifically tested in MF cohorts of patients, and further information is warranted.

One of the easiest tools for the evaluation of patients’ eligibility for a transplant is represented by Karnofsky performance status (KPS). On the basis of a simple scale from 0 (death) to 100 (normality), according to subject well-being, KPS has been invariably associated with transplant outcomes, including myeloablative and reduced intensity conditioning platforms. Patients with scores lower than 90 are generally projected to a worse transplant result, due to increased non-relapse mortality [32]. 

Then, comorbidity has been extensively considered in the past in order to properly select eligible candidates for a transplant. Probably, the most used tool for patients’ evaluation and selection is represented by the hematopoietic cell transplantation-specific comorbidity index (HCT-CI). Developed by Sorror et al. and published in 2005, this score includes the presence and severity of 15 comorbidities. HCT-CI classifies patients at low (0), intermediate (1–2), and high (≥3) risk, correlating with worse survival due to increased non-relapse mortality [33]. HCT-CI comorbidity score was developed and validated on cohorts of relatively young patients (median age <50 years) [34,35]. In addition, this score has not been extensively studied in MF; therefore, we actually do not know whether it can represent an effective tool for the selection of MF allo-HCT candidates. In fact, some reports have highlighted some peculiar aspects of elderly candidates for a transplant, as usually MF allo-HCT patients are. 

In this regard, as observed in other hematological diseases, the median age at transplantation progressively increased, as well as numbers of MF transplant procedures, over time [9]. In the early 2000s, only a small proportion of patients were submitted for a transplant after 60 years; later on, the dramatic advances in transplant procedure led to consider transplant in older adults, even up to 75 years. The utility of transplant over 70 years is questioned by several transplant physicians due to concerns related to NRM and effective survival benefit in such a population. Recently, a joint study from the Spanish MF registry and Chronic Malignancies Working Party of the EBMT addressed the role of transplant in MF patients over 65 years [36]. The authors reported, once again, a survival benefit for the transplant after the first year from the procedure; importantly, increasing a recipient’s age did not correlate with worse outcome, thus supporting the idea to consider transplant also in more advanced age groups, at least up to 75 years. For this category of patients, it is necessary to perform a comprehensive evaluation taking into account physiological and geriatric components. Several experiences documented the complexity of evaluating elderly fitness in onco-hematology [37]. In addition, some reports showed a poor predictive value of the original HCT-CI and suggested the implementation of geriatric components for improving patients’ stratification [38]. Recently, we published an experience on 228 elderly (>60 years) allo-HCT patients, including 18 patients with MF. In this cohort, a multidimensional geriatric assessment (FIL score) was found to highly predict NRM and, therefore, allo-HCT outcome. Thus, this tool might be set to become a new instrument for the selection of elderly candidates for a transplant, including MF patients [39]. Several other multidimensional scores are currently under investigation, and important information is awaited in the near future.

Finally, by combining disease-, patient-, and donor-specific features, a Myelofibrosis-specific Transplant Scoring System (MTSS) has been proposed [40]. On the basis of the following clinical and molecular variables (leukocytes >25 × 10^9^/L, platelets <150 × 10^9^/L, Karnofsky scale <90%, age >57 years, ASXL1 mutation (1 point each), JAK2-mutated or triple negative status (2 points), and mismatched unrelated donor (2 points)), patients are stratified in 4 different groups: low (score 0-2), intermediate (score 3–4), high (score 5), and very high (score >5), with a post-transplant 5-year survival estimation of 90%, 77%, 50%, and 34%, respectively. The performance of this score was found to be higher compared to all the other available tools.

In the effort to implement all disease- and transplant-specific scores, all MF patients up to 75 years with high risk disease (as defined as life expectancy lower than 5 years according to standard MF-oriented prognostic scores) should be considered possible candidates for a transplant. For those patients at low-intermediate-risk MTSS with no significant contraindications, transplant should be pursued as soon as possible; on the contrary, other treatment options should be evaluated in very high risk-risk MTSS category or with severe comorbidities. In a high MTSS risk group of patients, where a 5-year NRM of 36% after transplant is expected, allo-HCT procedure should be chosen on a case-by-case basis taking into account patient preference and other possibly relevant factors (comorbidities, cognitive status, geriatric assessment).

## 4. Donor Choice and Stem Cell Source

Donor source plays a crucial role in MF-patients transplanted outcome. Indeed, the use of HLA-mismatched unrelated donors have been reported to be an independent risk factor for both disease-free and overall survival in several reports [41,42,43]. On the other hand, there is a general consensus that, as for other diseases, an HLA-matched donor, either sibling or unrelated, is associated with a better outcome [40,44], although few authors claim that MF patients transplanted from an unrelated donor, regardless of HLA-matching status, present a worse survival [43]. 

Recently, alternative donors have been used, showing proof of their role in MF transplant [45,46,47]. The use of cord blood was reported only for a minority of patients. One of the major limitations for use of this source of stem cells is represented by a scarce number of stem cells, leading to increased risk of graft failure and higher NRM. In fact, previous experiences showed a remarkable 40% of engraftment failure among MF patients receiving cord blood transplant [48]. At the moment, in the opinion of the authors, this type of transplant still remains experimental in MF. Conversely, haploidentical donors have been increasingly employed over time. Evidence is increasing in favor of this option that seems to offer similar results compared to HLA-matched donors also in MF setting [45]. 

Taking into account stem cells source, there is no consensus on the preferred source of stem cells (bone marrow versus peripheral-derived stem cells), although PBSC seems to guarantee a faster recovery [47,49]. In addition, higher stem cells doses (>7 × 10^6^ CD34+ cells/Kg) can lead to a faster engraftment and superior survival, particularly in a sibling setting, as recently demonstrated [50,51].

In summary, when considering allo-HCT in MF, sibling donors and high stem cells doses should be considered as the best options. Matched-unrelated or haploidentical donors might be a second preferred choice.

Beside HLA-compatibility and stem cell source other parameters could be considered: donor age and gender, female parity, AB0 compatibility, and CMV serostatus combination represent other important points for discussion when more than one donor is available [52].

## 5. Conditioning Regimen

Unfortunately, there are limited data on the optimal conditioning regimen because of the lack of prospective clinical trials comparing myeloablative (MAC) to reduce insensitivity conditioning (RIC) regimens in MF. Patients’ comorbidity and functional status can significatively influence the choice of conditioning regimen. Indeed, retrospective studies in the pre-ruxolitinib era indicate that MAC should be preferred in young patients without comorbidities and with an HLA-matched sibling donor, while RIC may be preferred in patients older than 50 years [14]. 

MAC regimens potentially have a good rate of survival, ranging from 47 to 61% of OS at 5 years from transplant [53,54]. Usually, the conditioning regimen was based on busulfan plus cyclophosphamide and total body irradiation with or without cyclophosphamide, but transplant related mortality (NRM) and GvHD rates were high, especially in older individuals, ranging from 20 to 48% at 1 year. RIC has been increasingly used in MF in consideration of the older age of MF allo-HCT candidates. 

Historically, the first prospective EBMT multicenter phase II trial of RIC consisted of busulfan (10 mg/kg) orally (or equivalent IV dose) plus fludarabine (180 mg/m^2^) (FLU-BU) and in vivo T-cell depletion with anti-thymocyte globulin. This protocol resulted in low rates of primary graft failure and rapid hematologic recovery [41]. Another commonly used RIC regimen is based on Fludarabine 90 mg/m^2^, combined with melphalan 140 mg/m^2^ (FLU-MEL); this protocol has been compared in a retrospective study with the BU-FLU regimen, showing an increased early toxicity and NRM, but better disease control with superimposable long-term outcomes [55]. Subsequently, a randomized GITMO study comparing fludarabine in combination with busulfan 10 mg/kg i.v. or thiotepa 12 mg/kg failed to identify significant differences in terms of clinical outcome [56].

It is evident that a direct comparison of RIC and MAC regimens is extremely difficult. A large retrospective analysis of the EBMT including 2224 patients with MF, stratified according to conditioning intensity, showed no statistically significant difference in terms of engraftment, GvHD, NRM, and overall survival, while there was a trend toward a higher relapse rate in patients receiving the RIC regimen [57]. 

In recent years, the scientific community has shown an increasing interest in the use of double-alkylating conditioning regimen thiotepa-busulfan-fludarabine (TBF). Based on some retrospective studies, such a regimen has been documented to favorably affect transplant outcome, thanks to a faster donor engraftment and better disease control [58,59,60,61]. Prospective and randomized trials are warranted to confirm these preliminary results and to give conclusive information on the preferred conditioning regimen. 

Taken together, these results support the current EBMT/ELN consensus guidelines that suggest to tailor the conditioning regimen intensity on the basis of a patient’s fitness and disease status [14,62].

## 6. Splenomegaly Management

Splenomegaly is a frequent finding in MF. More than 80% of MF patients present splenomegaly at diagnosis, and, in a significant proportion of cases, the spleen has a considerable size, with around a quarter of patients presenting a spleen palpable more than 16 cm below left costal margin [11,63,64]. Biological and clinical studies support the pivotal role of spleen in MF disease maintenance and progression [65,66,67]. In addition, previous reports documented that spleen size and splenectomy before allo-HCT could affect engraftment and possibly survival [54,68]. In a recent European multicenter study reporting 546 patients with available information on spleen size at the time of transplant, patients undergoing a transplant with a spleen palpable below 5 cm from left costal margin presented the best transplant outcome compared to patients with a spleen between 5 and 14 and more than 15 cm, respectively. The increasing risk of death was found to be related to non-relapse mortality, with patients belonging to the lower category presenting a significantly shorter time to engraftment [69]. This report confirmed a prior observation by Bacigalupo et al., who reported a higher NRM in patients with splenomegaly diameter >22 cm by ultrascan evaluation [70]. 

It is, therefore, crucial to pay particular attention to splenomegaly management before transplant.

In the next paragraphs, we will discuss about treatment options for MF patients with significant splenomegaly.

### 6.1. Medical Options (JAK Inhibitors)

The availability of JAK inhibitors has changed the treatment paradigm of MF patients. Based on the efficacy in reducing constitutional symptoms and splenomegaly, ruxolitinib has become the first treatment option for patients complaining of disease-related manifestations [71,72,73,74]. 

Better patients’ conditions and spleen shrinking are expected to have a positive influence on allo-HCT, too. In addition, improvement of pro-inflammatory status typical of MF may favor a positive graft function [75], thus decreasing the risk of graft failure and poor graft function, a life-threatening complication frequently occurring after transplant in MF patients [62,76].

In 2016, Shanavas first reported on a quite large cohort of 100 allo-HCT MF patients with prior exposure to JAK-inhibitors (90% ruxolitinib-treated). Around one quarter of patients achieved a clinical improvement while on the treatment, and their transplant outcome was significantly better as compared to either non-responsive or progressive patients. As expected, the inferior survival in patients with blast-phase MF was due to higher relapse risk in this cohort; conversely, a better NRM was observed in JAKi-responsive compared to non-responsive patients [44]. Subsequently, Kroger et al. led a multicenter retrospective analysis on behalf of EBMT, collecting a cohort of 551 patients, of whom 277 received ruxolitinib prior to allo-HCT between 2012 and 2016. RUX-responsive patients presented a faster engraftment and lower risk of relapse with consequent better event-free survival, defined as the time from allo-HCT until relapse, disease progression, or death, whichever occurred first [77]. According to spleen size, those patients achieving a spleen reduction over 5 cm below left costal margin on ruxolitinib seemed to have the better outcome, suggesting to titrate ruxolitinib to the maximum tolerated dose in the effort to reach the lowest spleen size as possible at the time of allo-HCT [69].

All these data support the pre-transplant use of ruxolitinib, even though drug-suspension strategy is still not well-defined. Preliminary data from the literature confirmed the safety of ruxolitinib therapy prior allogeneic steam cell transplant (allo-HCT), although some unexpected side effects were recorded when ruxolitinib was abruptly discontinued [44,78,79,80]. The cytokines rebound after ruxolitinib withdrawal may have induced some cases of cardiac shock and tumor lysis syndrome recorded in those experiences [81]. These reports have induced researchers to extend the use of ruxolitinib to the conditioning regimen phase, or even until engraftment [82,83,84]. All the studies showed favorable transplant outcomes, with particularly reduced risk for GVHD, indirectly confirming the results of multicenter phase III REACH trials [85,86], where ruxolitinib also has proven activity in treating GVHD. Table 2 summarizes the available studies that have investigated the role of ruxolitinib prior to or peri-transplantation.

When ruxolitinib fails, second generation JAK-inhibitors can be considered. Recently, fedratinib has received FDA and EMA approval for its use in this setting [91,92]. 

In the JAKARTA-2 trial including MF patients with prior exposure to ruxolitinib, around 30% of subjects obtained a spleen response (≥35% spleen volume decreases at 6-month evaluation) and/or symptom response [93]. Side effects included hematology, gastrointestinal effect, and deficit in thiamine blood concentration, manageable with supportive treatments (prokinetics and anti-diarrheal drugs plus thiamine supplement).

Momelotinib and pacritinib are other possible options with interesting effectiveness spectrum. Momelotinib showed mild efficacy in spleen reduction after ruxolitinib exposure; however, a significant reduction of transfusion-independency (41%) was observed in phase III SIMPLIFY-2 trial [94]. Pacritinib documented efficacy (20–30% in spleen volume reduction and/or symptoms response) in the subset of patients with low-platelet counts (<100 × 10^9^/L), a somewhat neglected category of patients [95].

Novel classes of drugs (e.g., BH3-mimetics, CDK-6-inhibitors, BET-inhibitors, telomerases-inhibitors, and others), alone or in combination with JAK-inhibitors, have proven initial efficacy in MF, with relevant biological effects (reduction of BM fibrosis, molecular burden decrease) in a significant proportion of patients [96]. Unfortunately, almost all ongoing studies exclude transplant candidates; therefore, their safety and efficacy in this setting needs to be fully explored.

### 6.2. Splenectomy

The role of splenectomy in MF prior to transplant has been under debate for decades. Several reports showed a shorter time to engraftment after splenectomy [54,68,70,97]; however, until 2021, no clear benefit was demonstrated on overall survival. Moreover, a prospective EBMT trial, evaluating Fludarabine-Busulfan plus ATG reduced intensity conditioning regimen reported an increase in relapse risk after transplant in splenectomized patients [41].

However, some biological and clinical information support the potential benefit of splenectomy. From the biological point of view, it is known that MF spleens carry additional molecular and cytogenetical abnormalities compared to peripheral blood; leukemic evolution inside spleen has also been described [65,66,67]. Therefore, spleen removal is expected to have a disease-modifying effect.

From the clinical point of view, surgery is expected to ameliorate thrombocytopenia, portal hypertension, anemia, and spleen-related symptoms in a significant proportion of patients [98]. These effects may be translated in a reduced risk for graft failure and improved graft function; on the other side, a better pre-transplant performance status could lead to improved non-relapse mortality.

Based on these premises, a retrospective study including 1195 MF allo-HCT patients was conducted with the aim to give a conclusive answer to the question whether to perform splenectomy before transplantation could affect long term transplant outcome. In that experience, 202 (17%) patients were submitted to splenectomy prior to transplant. As expected, the proportion of surgical procedures tended to decrease over time, probably thanks to the availability of novel treatments (e.g., ruxolitinib). Splenectomy was confirmed to be associated to a faster neutrophil and platelet recovery and lower non-relapse mortality with an increased relapse risk.

However, in patients with progressive disease, splenectomy prior to transplant was found to have a positive effect. In fact, splenectomized patients had a 54% decrease in death risk compared to subjects with progressive splenomegaly over 15 cm below left costal margin. In this context, the excess of relapse was not evident [69].

Certainly, splenectomy in MF is burdened by a significant morbidity and mortality: thrombo-hemorrhagic events and infections are frequent. Overall, around 1/3 of MF patients undergoing surgery will experience a perioperative complication, and mortality is reported in about 5–10% of cases. Bleeding and thrombosis are prominent, and roughly 10–15% of patients can experience such complications [98]. Among the recognized risk factors for vascular events, spleen mass, leukocytosis, and thrombocytopenia play a significant role. In those patients with thrombocytosis and or leukocytosis, cytoreductive treatment might prevent thrombo-hemorrhagic complications and should be taken into account. Early mobilization and use of anti-coagulant can be also useful.

In addition, anti-infective prophylaxis is mandatory. Post-splenectomy infections are frequent, with around 10% of subjects experiencing this complication; overwhelming post-splenectomy infection (OPSI), often associated to encapsulated bacteria, is a serious concern. Vaccination before splenectomy is, therefore, a mainstay. Pneumococcal, meningococcal, and H.influenzae vaccinations are highly recommended before splenectomy, ideally >2 weeks before a planned splenectomy. In adults, no clear evidence supports the routine use of antibiotics as primary prophylaxis; however, in some selected high-risk cases (patients with infectious history, etc.), penicillin or alternatives can be given [99].

Furthermore, a higher risk of leukemic transformation has been reported, by propensity score analysis, in an Italian multicenter retrospective study. In that experience, a higher than two-fold increase in blast evolution was observed among splenectomized subjects [100]. It should be noted, however, that prognostically-relevant cytogenetical and molecular analyses were not available at that time. Therefore, caution is needed in data interpretation. For this concern, it should be considered to proceed early to transplantation after splenectomy, ideally within 1–3 months, if feasible.

Can all these complications preclude or delay allo-HCT? With the aim to answer this relevant question, a French nationwide study was conducted [101]. In this trial, all MF patients undergoing unrelated donor search in the French national registry during the period 2008–2016 were recruited. Patients who had received splenectomy before registration in French registry were excluded. Patients were followed from the registration to death, splenectomy, lost to tracking, or the end of the study for at least 18 months. By applying a multistate model, the researchers documented that splenectomy was significantly associated with higher probability of transplant within 4 months after surgery. Importantly, no increase in death risk was observed after splenectomy and only few splenectomy-related complications were indicated as the leading cause of transplant preclusion.

The current evidence, therefore, suggests the use of splenectomy in all suitable patients with progressive splenomegaly, while on any medical treatment, palpable over 15 cm below left costal margin.

### 6.3. Splenic Irradiation

Splenic irradiation (SI) has been used in the past for splenomegaly-related symptoms in those patients with high surgical risk, particularly before the advent of JAK-inhibitors. 

Radiation dosage widely varies among Centers, usually ranging from 100 to 1000 cGy in 5–10 fractions. The mechanism of action of radiation therapy on MF spleen is still fairly unknown. SI is thought to have a role in reducing the number of neoplastic cells into the spleen, leading to improvements in both splenic size and discomfort. Unfortunately, the benefit of splenic irradiation is usually short-lived, and worsening of cytopenias is frequently observed.

Overall, information on SI relies on small retrospective single-center experiences. In addition, scant information about the its role before transplant is available.

In the largest study available, reporting 23 MF patients submitted to SI due to symptomatic splenomegaly, the response rate on splenomegaly was 93.9%, with a median maximal decrease in spleen length of 5 cm. Symptom relief was documented in 95.6% of patients; however, the benefit duration was limited to a median of 6 months. Significant cytopenias were recorded in around one half of the study cohort, with life-threatening events in 26% of patients [102]. 

The effects of splenic irradiation prior to transplant have been investigated in small case series, with proof of mild efficacy [103]. Currently, a multicenter study on behalf of Chronic Malignancies Working Party of EBMT is ongoing to elucidate the role for this procedure before or during allo-HCT. To date, splenic irradiation could be offered to those MF patients with massive splenomegaly and surgical contraindications only in experienced Centers.

## 7. Management of Cytopenia before Transplant

Another important issue to be managed before transplant is the presence of cytopenias. Indeed, cytopenias are frequent in patients with MF: significant anemia is recorded in around 1/3 of patients at diagnosis, and its worsening with the requirement of red-blood cells support is one of the strongest predictors of survival [11,12]. Similarly, thrombocytopenia may be associated to increased risk of leukemic transformation [104]. MF-directed treatment, namely JAK-inhibitors, may worsen cytopenias itself due to on- and off-target mechanism of action; as a result, dose reduction or discontinuation is frequently required, leading to disease unresponsiveness or progression [7,105].

One of the most relevant effects of transfusion-dependency is represented by iron overload. Ineffective erythropoiesis may aggravate iron accumulation. A bulk of literature highlights the negative effect of iron for organs, such as liver, heart, joints, and endocrine organs. Hepato-cirrhosis, cardiomyopathy, and several endocrinologic disturbances, caused by increased production of reactive oxygen species, have been documented in long-lasting iron overload [106]. Iron overload may be even more important in allo-HCT recipients. The excess of iron may induce short and long-term complications [107]. Infectious risk from opportunistic agents (such as fungi and bacteria), acute and chronic GHVD, and sinusoidal obstruction syndrome are more frequently observed in such patients [108]. Importantly, iron overload may affect bone marrow microenvironment leading to a not-permissive habitat for hematopoietic stem cells with increased risk for graft failure and poor graft function, particularly fearsome in the MF setting [62,109,110,111]. In addition, a high number of pre-transplant blood units did correlate with unfavorable prognosis in an Italian study by Bacigalupo et al. [70].

Early intervention should be advisable, particularly in patients with a hemoglobin value lower than 10 g/dL. Available drugs for the management of anemia in MF include corticosteroids, androgens, danazol, immunomodulating agents (e.g., thalidomide, lenalidomide), and epoetins.

Single-agent corticosteroid (prednisone 0.5 to 1 mg/Kg/day) and androgen therapy (e.g., testosterone enanthate 400 to 600 mg weekly, oral fluoxymesterone 10 mg three times per day, or danazol at a dose of 600 mg/day) have been used in MF cases, obtaining a response rate ranging from 30 to 40% [112,113,114]. Low dose thalidomide (50 mg/day) as single agent or in combination with corticosteroids (prednisone 15 to 30 mg/day) and lenalidomide (5–10 mg/d), in the presence of del(5)(q31), showed a response rate of approximately 20% [115,116,117].

Immunomodulating agents have also been associated with responses on thrombocytopenia [118,119].

Keeping in mind the possible side effects of these drugs, androgen preparations should be avoided in patients with prostate disease or concomitant liver disease, thalidomide and its analogs should be used with caution in patients with or at risk of thrombosis and neuropathy, and corticosteroids may significantly increase infectious risk or metabolic disturbances.

Response of anemia to epoetin treatment has been reported in 45–50% of MF patients, mainly in the context of inadequate endogenous erythropoietin level (<125 U/L) and non-transfusion-dependent anemia: female sex, leukocyte count ≥10 × 10^9^/L, and serum ferritin < 200 ng/mL seem to confer a significantly higher probability of response to erythropoiesis-stimulating agents [120,121,122]. The efficacy of epoetin treatment was also documented among ruxolitinib-treated patients [123].

Finally, there are new promising agents under evaluation: the activin receptor ligand traps (e.g., luspatercept), currently approved for refractory anemia with ring sideroblasts (RARS-MDS); recombinant pentraxin-PRM-151 with anti-fibrotic activity; and new generation JAK-inhibitors (momelotinib, pacritinib), as discussed in the previous chapter. The present available data about these new drugs come mainly from phase II trials, and more information is required to understand their real clinical benefit in MF.

In those patients with transfusion-dependent anemia and iron overload, iron chelation therapy is the main pharmacological option for decreasing iron deposits, possibly reducing the risk of short- and long-term post-allo-HCT complications. While deferoxamine (short-half-life and prolonged infusion) and deferiprone (risk for agranulocytosis) have a minimal role in chelation of such patients, deferasirox, an oral iron-chelator, is increasingly used. This drug has proven its efficacy in reducing iron burden with manageable toxicity (mainly creatinine increase) in transfusion-dependent patients [124].

Its efficacy has been also confirmed in the MF context. Deferasirox is able to achieve a significant response in terms of chelation and anemia improvement, particularly when started early [125,126]. The use of deferasirox chelation therapy in the context of allo-HCT has been evaluated in several prospective and retrospective studies [107]. Unfortunately, none of those specifically focused on MF. The large majority reported favorable effect on chelation and transplant outcome. Taking into account the deleterious effect of iron toxicity on engraftment [127], it seems appropriate to limit iron overload prior to transplant in MF. Post-transplant chelation is another possible option, even though the concomitant use of nephrotoxic agents (calcineurin inhibitors, antivirals, etc.) may limit its feasibility, particularly in the peri-transplant period.

These data encourage the timely use of deferasirox in those patients with transfusion-dependent anemia and signs of iron overload. Ideally, all patients receiving support with more than 10 units of blood and/or serum ferritin >1000 ng/mL should receive iron-directed treatment. Allo-HCT should be delivered as soon as possible to such patients in order to limit the number of transfusions and the excess of iron.

## 8. Timing of Transplant

Last, timing of transplant in MF is of paramount importance. The availability of novel and effective drugs has led many MF experts to consider a delay of transplant at the loss of response. Whether to proceed to transplant early or after treatment failure is a controversial area. At the same time, some data support the idea of not waiting for the transplant at the time of disease progression. The first item supporting this approach is that around 70% of MF patients receiving JAK inhibitors are expected to discontinue treatment at 5-year follow-up [128]. As described before, a progressive splenomegaly may induce a delayed engraftment and increased risk of non-relapse mortality after transplant [69]. Second, it is well recognized that JAK-inhibitors do not affect the risk of leukemic evolution [129]. For patients evolving in accelerated or blastic phase of MF, the prognosis is poorer. A 2-fold increase in relapse incidence was found among patients submitted to transplant with accelerated phase of disease (blast cells 10–19%) [130]. In the overt blast phase, the probability to achieve a long-term disease control after transplant is severely reduced, even in the context of a pre-transplant complete remission [131].

As a consequence, our approach is to proceed to the transplant as soon as possible, given the time of best disease response.

## 9. Conclusions

This critical review highlights the complexity and the importance of pre-transplant management of MF (Figure 1). Many small details can make a big difference in MF transplant outcome. As a consequence, a strict collaboration between MF and allo-HCT physicians should be pursued for laying the foundations of cures for MF patients.

## Figures and Tables

**Figure 1 cells-11-00553-f001:**
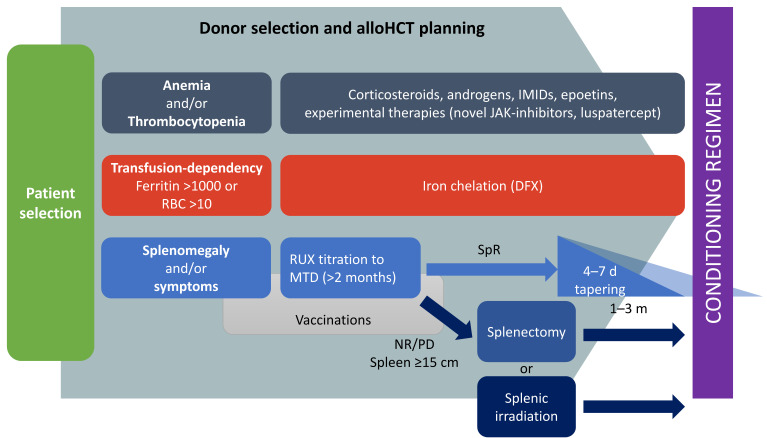
Proposed pre-transplant management of MF candidates to allogeneic stem cells transplantation. IMIDs: immunomodulating agents; RUX: ruxolitinib; MTD: maximum tolerated dose; RBC: red blood cells; SpR: spleen response; NR: no response; PD: progressive disease.

**Table 1 cells-11-00553-t001:** List of the available risk score for MF and their prognostic relevance. PMF: Primary Myelofibrosis; sMF: Secondary Myelofibrosis; HMR: high molecular risk; Int: intermediate; TD transfusion-dependent; BM: bone marrow; OS: overall survival.

Risk Score		IPSS	DIPSS	DIPSS-Plus	MYSEC-PM	MIPSS70	MIPSS70-Plus v2.0	GIPSS
Applicability		PMF at diagnosis	PMF at any time	PMF at any time	sMF at diagnosis	PMF at any time	PMF at any time	PMF at any time
Features		Clinical	Clinical	Clinical & Molecular	Clinical & molecular	Clinical & molecular	Clinical & molecular	Genetical only
Items (points)	Age	>65 y (1)	>65 y (1)	>65 y (1)	Age (0.15/y)	–	–	–
	Leucocytes	>25 × 10^9^/L (1)	>25 × 10^9^/L (1)	>25 × 10^9^/L (1)	–	>25 × 10^9^/L (1)	–	–
	Blasts	≥1% (1)	≥1% (1)	≥1% (1)	≥3% (2)	≥2% (2)	≥2% (2)	–
	Constitutional symptoms	Yes/No (1)	Yes/No (1)	Yes/No (1)	Yes/No (1)	Yes/No (1)	Yes/No (2)	–
	Hemoglobin	<10 g/dL (1)	<10 g/dL (2)	<10 g/dL (2)	<11 g/dL (2)	<10 g/dL (1)	<8 g/dL (F)/<9 g/dL (M) (2) 8–9.9 g/dL (F)/9–10.9 g/dL (M) (1)	–
	TD-anemia	–	–	Yes/No (1)	–	–	–	–
	Cytogenetics	–	–	Unfavorable ^£^ (1)	–	–	very high risk ^§^ (4) unfavorable ^§^ (3)	very high risk ^§^ (2) unfavorable ^§^ (1)
	Platelets	–	–	<100 × 10^9^/L (1)	<150 × 10^9^/L (2)	<100 × 10^9^/L (2)	–	–
	Molecular	–	–	–	No CALR (2)	No CALR type-1 (1) HMR ^%^ mutation (1) >1 HMR ^%^ mutations (2)	No CALR type-1 (2) HMR ^$^ mutation (2) >1 HMR ^$^ mutations (3)	No CALR type-1 (1) ASXL1 (1) SRSF2 (1) U2AF1Q (1)
	BM fibrosis	–	–	–	–	Grade ≥2 (1)	–	–
Higher risk Categories (score) Median OS		Int-2 (2): 4 y High (3–4): 2.3 y	Int-2 (3–4): 4 y High (5–6): 1.5 y	Int-2 (2–3): 2.9 y High (4–6): 1.3 y	Int-2 (>14 < 16): 4.4 y High (≥16): 2 y	Int (2–4): 7.1 y High (>4): 2.3 y	High (5–8): 4.1 y Very high (≥9): 1.8 y	Int-2 (2): 4.2 y High (≥3): 2 y

^£^ Unfavorable karyotype: complex karyotype or single or two abnormalities, including +8, −7/7q-, i(17q), −5/5q-, 12p-, inv(3), or 11q23 rearrangement. ^%^ HMR mutations according to MIPSS70: ASXL1, SRSF2, EZH2, IDH1, IDH2. ^$^ HMR mutations according to MIPSS70-plus v2.0: ASXL1, SRSF2, EZH2, IDH1, IDH2, and U2AF1Q157. ^§^ Very unfavorable karyotype: single/multiple abnormalities of −7, i(17q), inv(3)/3q21, 12p-/12p11.2, 11q-/11q23, or other autosomal trisomies, not including +8/+9 (e.g., +21, +19); Favorable: normal karyotype or sole abnormalities of 13q-, +9, 20q-, chromosome 1 translocation/duplication or sex chromosome abnormality, including -Y; ‘Unfavorable‘: all other abnormalities.

**Table 2 cells-11-00553-t002:** List of studies investigating the role of ruxolitinib (RUX) in the pre- or peri-transplant period.

Author	Year	N	Study Design	Conditioning Regimens	RUX Use	Spleen Response	RUX Tapering Strategy	Discontinuation Syndrome	GF (%)	G2-4 aGVHD (%)	NRM (%)	OS (%)
Jaekel, N. [78]	2014	14	Retro	RIC (Flu-Bu/TBI) MAC (NA)	Pre	64%	Stop at conditioning	None	7%	14%	7% at 9 m	50% at 1 y
Stübig, T. [79]	2014	22	Retro	RIC (Flu-Bu/Mel/Treo)	Pre	45% (>50%) 24% (<50%)	Stop at conditioning	None	0%	36%	14% at 1 m	81% at 1 y
Shanavas, M. [44]	2016	100	Retro	RIC (Flu-Bu/Mel/Cy/BCNU/TBI) MAC (Flu-Bu/Mel or Bu-Cy)	Pre	23%	Not defined	10%	8%	37%	28% at 2 y	61% at 2 y
Kroger, N. [84]	2018	12	Prosp	RIC (Flu-Bu)	Peri	100%	Stop at day +28 post-transplant	None	0%	8%	0% at 17 m	100% at 17 m
Kadir, S.S.S.A. [87]	2018	46	Retro	RIC (Flu-Bu/ FLAMSA-Flu-Bu)	Pre	39%	Not defined	None	4%	37%	23% at 2 y	73% at 2 y
Gupta, V. [88]	2019	21	Prosp	RIC (Flu-Bu)	Pre	45%	Tapering off at conditioning (4 days before)	None	16%	47%	28% at 2 y	66% at 2 y
Salit, R.B. [82]	2020	28	Prosp	RIC (Flu-Mel) MAC (Bu-Cy±Flu)	Pre	NA	During Conditioning (day-4)	None	0%	78%	7% at 13 m	86% at 2 y
Ali, H. [89]	2021	18	Prosp	RIC (Flu-Mel)	Peri	NA	Day +30 post-transplant	None	0%	17%	23% at 1 y	77% at 1 y
Kroger, N. [77]	2021	277	Retro	RIC (NA) MAC (NA)	Pre	56%	Not defined	6%	NR	29%	26% NR at 1 y 15% R at 1 y	66% at 2 y
Robin, M. [90]	2021	59	Prosp	RIC (Flu-Mel)	Pre	46%	Variable	15.8%	3%	66%	42% at 1 y	68% at 1 y

Retro: retrospective; Prosp: prospective; NA: not available; Flu: fludarabine; Bu: Busulfan; TBI: total body irradiation; Cy: cyclophosphamide; MAC: myeloablative conditioning; RIC: reduced intensity conditioning; GF: graft failure; aGVHD: acute graft versus host disease; NRM: non-relapse mortality; OS: overall survival; NR: no response; R: response.

## Data Availability

Not applicable.

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
