# Peer review of "How We Manage Myelofibrosis Candidates for Allogeneic Stem Cell Transplantation"

_cells, 2022, doi:10.3390/cells11030553_

Round 1

Reviewer 1 Report

The manuscript is well-written and addresses a clinical important question.  The management of myelofibrosis patients being candidates for allogeneic HSCT is often associated with significant clinical difficulties and the review addresses these problems in a very comprehensive way.

I have just some minor comments.

  • Line 122: The paragraph adressing the question of how to deal with patients that have marked differences in their prognostic scores is not finished. It cleary poses the question, but does not really provide an answer. The presumable answer is only exposed in the next paraphragh summarizing the opinions. Consider rephrasing this section to make it more clear.
  • The table 1 with the different scores is not very clear. I would suggest to put the different risk factors in the left column to better evidence differences between the scores. Furthermore, the table needs some editing for the column alignment
  • line 163: high risk: There is an error of the special sign (ε3).
  • Line 328-330 and 474: Different font from the rest of the manuscript
  • Table 2: Consider also to include the different MAC and RIC conditionings used in the displayed studies.
  • The figure 1 showing the management of the patients in the pretransplant setting should also include the management of cytopenias as described in the text.

Author Response

The manuscript is well-written and addresses a clinical important question.  The management of myelofibrosis patients being candidates for allogeneic HSCT is often associated with significant clinical difficulties and the review addresses these problems in a very comprehensive way.

I have just some minor comments.

  • Line 122: The paragraph adressing the question of how to deal with patients that have marked differences in their prognostic scores is not finished. It cleary poses the question, but does not really provide an answer. The presumable answer is only exposed in the next paraphragh summarizing the opinions. Consider rephrasing this section to make it more clear.

Ok. The paragraph has been modified according to your suggestion (lines 151-162)

  • The table 1 with the different scores is not very clear. I would suggest to put the different risk factors in the left column to better evidence differences between the scores. Furthermore, the table needs some editing for the column alignment

Ok. Table 1 has been revised taking into account your comment.

  • line 163: high risk: There is an error of the special sign (ε3).

Ok. This typo has been corrected (line 213).

  • Line 328-330 and 474: Different font from the rest of the manuscript

Ok. The text has been standardized in font throughout the manuscript

  • Table 2: Consider also to include the different MAC and RIC conditionings used in the displayed studies.

Ok. Table 2 was modified in order to include different MAC and RIC conditioning, whenever possible

  • The figure 1 showing the management of the patients in the pretransplant setting should also include the management of cytopenias as described in the text.

Ok. Management of cytopenias have been included in figure 1.

Reviewer 2 Report

This scoping review by Polverelli et al covers the indications, donor selection, condition regimen, and outcomes of allogeneic transplantation in myelofibrosis. Although this manuscript is missing several key topics, overall the quality and breadth of discussion is a benefit to the literature and physicians caring for MF patients. See specific comments below

Major:

  • The discussion of transplant indication is lackluster with respect to data for intermediate-1 patients. Please include a more thorough discussion of the Kroger et al study (PMID 25784679) and the more recent Gowin et al CIBMTR study (PMID 32384540).
  • The patient selection section would benefit from a more thorough discussion of age cutoff, specifically outcomes of transplantation in elderly MF patients.
  • Can the authors comment on the utility, or lack of utility for umbilical cord blood donation in MF?
  • Splenectomy is being suggested for all "all suitable patients with progressive splenomegaly palpable over 15 cm below left costal margin." This sentence should also qualify that splenectomy should only be considered in the setting of JAKi refractory disease, as suggested by the authors prior study (PMID 33064301)
  • The iron chelation section should include specific data for these agents prior to transplantation in both MF and other myeloid disease. Many centers do not chelate prior to transplantation and instead favor post transplantation chelation. Therefore, the authors need to provide support for their recommendation for pre-transplant chelation.
  • There is a paucity of discussion regarding the optimal timing of transplantation in MF. In stable patients, upfront versus delayed transplantation should be discussed. The authors should also include a brief discussion of outcomes of MPN-AP (Gagelmann ASH 2021) and MPN-BP (Gupta Blood Advanced 2020) as a way to support transplantation before disease transformation. 

Minor:

  • Page 1, line 28: Likely a formatting error, but the first five paragraphs can be folded into one
  • Page 3, line 103: The MPD-RC study is not cited. The PMID is 30625392
  • Page 4, line 163: Should be >3, formatting error
  • There is inconsistent use of abbreviations throughout the manuscript, specifically MF
  • The references start over after 119, another formatting error 

Author Response

This scoping review by Polverelli et al covers the indications, donor selection, condition regimen, and outcomes of allogeneic transplantation in myelofibrosis. Although this manuscript is missing several key topics, overall the quality and breadth of discussion is a benefit to the literature and physicians caring for MF patients. See specific comments below

Major:

  • The discussion of transplant indication is lackluster with respect to data for intermediate-1 patients. Please include a more thorough discussion of the Kroger et al study (PMID 25784679) and the more recent Gowin et al CIBMTR study (PMID 32384540).

Thanks for your comment. The discussion regarding transplant indication for intermediate-1 patients has been integrated. The two studies have been cited accordingly (lines 88-99).

  • The patient selection section would benefit from a more thorough discussion of age cutoff, specifically outcomes of transplantation in elderly MF patients.

Ok. Patient selection chapter has been revised in order to discuss more specifically transplantation in elderly MF patients (lines 220-233, line 262).

  • Can the authors comment on the utility, or lack of utility for umbilical cord blood donation in MF?

Ok. The role of cord blood donation has been more extensively discussed (lines 280-285)

  • Splenectomy is being suggested for all "all suitable patients with progressive splenomegaly palpable over 15 cm below left costal margin." This sentence should also qualify that splenectomy should only be considered in the setting of JAKi refractory disease, as suggested by the authors prior study (PMID 33064301)

Thanks for your comment. The prior study documented a benefit for splenectomy in the category of progressive patients. Based on the retrospective nature of that reports, the authors were not aware whether the patients progressed during JAK-inhibition or another medical approach. Therefore, the suggestion for splenectomy can be extended to all progressive patients receiving any medical therapy. This has been clarified in the text (line 658).

  • The iron chelation section should include specific data for these agents prior to transplantation in both MF and other myeloid disease. Many centers do not chelate prior to transplantation and instead favor post transplantation chelation. Therefore, the authors need to provide support for their recommendation for pre-transplant chelation.

The authors agree with the reviewer. Post-transplant chelation is a feasible option; nevertheless, we believe that a significant risk for graft failure in such a setting should induce MF allo-HCT candidates to initiate iron chelation already prior to transplant. This is supported by some previous studies showing delayed engraftment due to iron toxicity. Also, the concomitant use of nephrotoxic agents (calcineurin inhibitors, antivirals, etc.) might reduce the feasibility of a post-transplant iron chelation. The text has been modified according to your suggestion. (lines 795-802)

  • There is a paucity of discussion regarding the optimal timing of transplantation in MF. In stable patients, upfront versus delayed transplantation should be discussed. The authors should also include a brief discussion of outcomes of MPN-AP (Gagelmann ASH 2021) and MPN-BP (Gupta Blood Advanced 2020) as a way to support transplantation before disease transformation. 

We totally agree with you. Timing of transplantation is matter of frequent debate among MF experts. Taking into account your evaluable suggestion, we added a paragraph in order to support an early recourse to transplantation (lines 809-825)

Minor:

  • Page 1, line 28: Likely a formatting error, but the first five paragraphs can be folded into one

Ok. This has been fixed (Iines 28-38)

  • Page 3, line 103: The MPD-RC study is not cited. The PMID is 30625392

Ok. The reference has been added (line 135)

  • Page 4, line 163: Should be >3, formatting error

Ok. This error has been fixed (line 213).

  • There is inconsistent use of abbreviations throughout the manuscript, specifically MF

Ok. Abbreviations have been used throughout the manuscript.

  • The references start over after 119, another formatting error 

Ok. References’ numeration has been corrected.